# The *C. difficile* toxin B membrane translocation machinery is an evolutionarily conserved protein delivery apparatus

Kathleen E. Orrell [1,2], Michael J. Mansfield[3], Andrew C. Doxey [3] & Roman A. Melnyk [1,2]*

Large Clostridial Toxins (LCTs) are a family of six homologous protein toxins that are implicated in severe disease. LCTs infiltrate host cells using a translocation domain (LCT-T) that contains both cell-surface receptor binding sites and a membrane translocation apparatus. Despite much effort, LCT translocation remains poorly understood. Here we report the identification of 1104 LCT-T homologs, with 769 proteins from bacteria outside of clostridia. Sequences are widely distributed in pathogenic and host-associated species, in a variety of contexts and architectures. Consistent with these homologs being functional toxins, we show that a distant LCT-T homolog from *Serratia marcescens* acts as a pH-dependent translocase to deliver its effector into host cells. Based on evolutionary footprinting of LCT-T homologs, we further define an evolutionarily conserved translocase region that we show is an autonomous translocase capable of delivering heterologous cargo into host cells. Our work uncovers a broad class of translocating toxins and provides insights into LCT translocation.

[1] Molecular Medicine Program, The Hospital for Sick Children Research Institute, Toronto, ON M5G 0A4, Canada. [2] Department of Biochemistry, University of Toronto, Toronto, ON M5S 1A8, Canada. [3] Department of Biology, University of Waterloo, 200 University Ave. West, Waterloo, ON N2L 3G1, Canada. *email: roman.melnyk@sickkids.ca

Large Clostridial Toxins (LCTs) are a family of bacterial toxins comprised of six proteins (TcdA, TcdB, TcsL, TcsH, TpeL, and TcnA)[1,2], defined first by their similar biochemical, immunological and pharmacological effects[3], and later differentiated by their clinical phenotype. TcdA and TcdB are the major causative agents of *C. difficile* infection, the leading cause of hospital-acquired diarrhea in developed countries[4], while other LCTs are implicated in gas gangrene, enterocolitis and toxic shock syndrome[5–8]. Although LCTs vary in their clinical manifestation, they all have highly similar structure and function. LCTs are high molecular weight (>200 kDa) single-chain polypeptides, sharing between 36 and 90% sequence identity[1] and inactivate GTPases in the Ras superfamily by glycosylation[9]. In order to gain entry into cells and access cytosolic GTPases, LCTs utilize their multi-domain architecture[10], much like other AB toxin families, including diphtheria toxin (DT)[11] and botulinum neurotoxin (BoNT)[12]. In brief, using their central translocation and receptor-binding domain (herein referred to as T-domain), LCTs bind cell-surface receptors and undergo receptor-mediated endocytosis. Low-pH mediated conformational changes in acidified vesicles culminates in insertion of regions of the T-domain into the endosomal membrane, resulting in formation of a translocation pore. The translocation pore facilitates passage of the LCT glycosyltransferase (GTD) and cysteine protease (CPD) into the cytosol, where the GTD is proteolytically released.

While much is known about the enzymatically active LCT domains, the function(s) of the LCT T-domain have remained much more elusive[10]. The LCT T-domain is much larger than the T-domain of other similar toxins (LCT: >100 kDa[13]; BoNT: ~50 kDa[12]; DT: ~20 kDa[11]), and has a unique structural fold at high pH[13]. The LCT T-domain at high pH is mostly composed of extended β-sheets, with a hydrophobic α-helical region that extends and wraps around the β-sheet structures. Within the β-sheet enriched region of the T-domain, four different LCT receptors have been identified (TcdB: CSPG4[14], Fzd[15,16], PVRL3[17]; TcdA: LDLR[18]; TpeL: LRP1[19]) that all bind within the C-terminal region of the T-domain, with one receptor (CSPG4) binding partially to the C-terminal repeating region (CROPS) of TcdB[20]. The dual functionality of the LCT T-domain to bind receptors and facilitate translocation has made it difficult to disentangle receptor-binding from translocation, although several studies have concluded that the N-terminal region of the T-domain is important for pore formation and translocation. We and others have identified a pore-forming region between residues 956–1115[21–23], which maps to the hydrophobic α-helical stretch in the T-domain, and important pore formation and translocation residues clustered between residues 1035–1107[22]. Recently, the structure of full-length TcdB was solved at endosomal pH with 3 neutralizing VHHs[24]. Conformational changes can be observed within the pore-forming region, although binding of a VHH within the pore-forming region and lack of a membrane prevent a complete understanding of the toxin structure at low pH and in the membrane. Outside 956–1115, the functional significance of the N-terminal region of the T-domain remains unclear. Comparison of the six LCT T-domain sequences does not reveal any striking patterns in conservation or hydropathy, and by extension, obvious clues into important functional regions[10].

In the past 5 years, genomics-driven approaches have facilitated the discovery of hundreds of bacterial toxin homologs, providing fundamental insights into toxin evolution and diversity[25]. Although homologs of major AB-toxins such as BoNT[26–29], DT[30], and others[31,32] have been identified using bioinformatic approaches, there have been no genomics-driven approaches to uncover and characterize LCT homologs. For BoNT and DT, most studies have focused on identification and characterization of homologs conserving the full toxin architecture, such as the BoNT-like toxin in a commensal strain of *Enterococcus faecium*[27,28]. Among multi-domain homologs, the receptor (which in other toxins is distinct from the T-domain) and effector domains are the most extensively analyzed, since these domains contain well-characterized functionally important residues. Compared to receptor and effector domains, less is known about AB toxin translocation, and to date, no study has used genomics to elucidate AB toxin translocation. Since homologs have been critical to understand the function of countless other proteins, we contend that T-domain homologs have the potential to make significant strides in our understanding of toxin translocation. Understanding the process of translocation is not only critical for a complete understanding of toxin entry and uptake into cells, but also has numerous applications, both in therapeutic interventions of toxin-mediated diseases, and in biotechnology applications, such as bacterial toxin-mediated drug delivery[33–35].

Here, we take a genomics-driven approach to uncover distant LCT-T homologs. We report hundreds of LCT-T homologs that are present in pathogenic and host-associated bacterial species outside of clostridia. We characterize an LCT-T homolog from *Serratia marcescens* and show that it causes changes in cell morphology and has pH-dependent translocation activity. We also leverage the LCT-T homologs to define an evolutionarily conserved translocation apparatus, which is present in all homologs. In addition to uncovering hundreds of toxins, which are linked through a shared mechanism of protein delivery into host cells, our results provide fundamental insights into translocation of the medically relevant LCTs.

## Results

**Identification of LCT-T homologs outside of clostridia.** To begin to explore the distribution, diversity, and function of the LCT T-domain, we searched 200,270 available genomes (8141 eukaryotes, 192,129 prokaryotes) within the Genbank database, and retrieved all sequences containing an LCT-like T-domain. To this end, a PSI-BLAST search was performed using the putative TcdB T-domain as a query (UniProt ID P18177.3, residues 800–1814). After removing partial and truncated hits 1,104 sequences were uncovered, including 335 LCT sequences found in various clostridia (*C. difficile*, *C. perfringens*, *C. novyi*, and *Paeniclostridium sordellii*, previously *C. sordellii*), and 769 sequences in species outside of clostridia (hereafter referred to as LCT-T homologs) (Fig. 1a). Similar PSI-BLAST searches of the glucosyltransferase, cysteine peptidase, and CROPS domains (UniprotKB P18177, residues 1–565, 567–774, and 1815–2361, respectively) yielded larger numbers of target sequences (5097, 3339, and >20,000 hits, respectively), implying that the translocase domain has a more restricted distribution among bacteria, potentially reflecting that its function is more specific or unique to LCTs. The translocase-related region of LCT-T homologs shares an average of 18.6% amino acid identity with the TcdB translocase, reflecting remote homology (Supplementary Fig. 1), although shuffled sequence comparisons to TcdB retain significant $E$-values ($E < 1e{-}5$) for 1023 of the 1104 sequences. LCT-T homologs are mostly distributed among the class Gammaproteobacteria (688 sequences across 32 genera); of the 32 genera, sequences are most common in *Pseudomonas* (419 sequences), followed by *Vibrio* (72 sequences), and *Providencia* (67 sequences) (Fig. 1b). The patchy distribution of LCT-T homolog sequences across the bacterial tree of life indicates that evolution by lateral gene transfer has likely played a strong role in the family's evolution, a pattern that has been observed in other toxin families (Supplementary Fig. 2)[36].

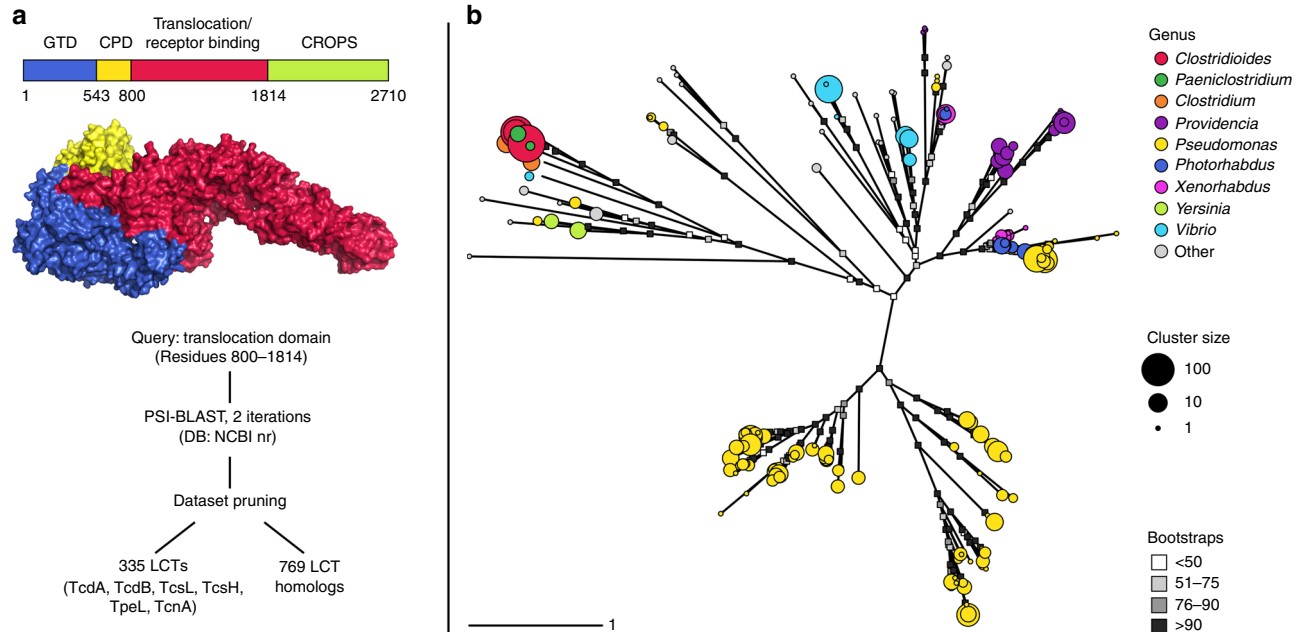

**Fig. 1 Discovery of LCT-like translocases in diverse species. a** The search strategy for discovering proteins with LCT-like translocases began by searching the NCBI non-redundant protein database (NR) using the TcdB T-domain as a query (UniProt P18177.3), followed by two iterations of PSI-BLAST searches. After removal of partial and poorly aligning sequences, a total of 1104 LCT and LCT-T homologs were retrieved. **b** A redundancy-removed and pruned alignment of translocase sequences was used to generate a maximum likelihood phylogeny. Each tip represents a sequence cluster centroid and colored according to their genus (genera that represent more than 1% of the total dataset are colored. Tip radius is proportional to the size of the cluster.

**LCT-T homologs have toxin-like signatures**. Consistent with their taxonomic distribution, analysis of metagenomes revealed LCT-T homologs in a broad distribution of environments (Supplementary Fig. 3). We detected LCT-T homologs in human gut, soil, wastewater, marine and aquatic environments, where the T-domains of other AB toxins (i.e. BoNT, DT) were conspicuously absent. Notably, LCT-T homologs were not encoded within analogous LCT pathogenicity loci (PaLoc)[37], with genes for toxin regulation (*tcdR, tcdC*) or toxin export (*tcdE*) (Supplementary Fig. 4). Despite their occurrence in a wide variety of genomic contexts, many LCT-T homolog genes were located near components of type I, III, IV, and VI secretion systems, and through proteomic association, the top co-occurring protein families with the LCT-T domain included secretion systems, along with many other virulence genes and mobile genes, including transposons and insertion elements (Supplementary Table 1). Together with their phylogenetic distribution, these data suggest that LCT-T homologs may function as putative toxins, many of which utilize non-LCT-modes of bacterial secretion and export.

**LCT-T homologs are found in pathogenic bacterial species**. In line with their toxin-like genomic signatures, 75% of LCT-T containing proteins were found in organisms with evidence of pathogenicity (Supplementary Fig. 5). In addition to species with known pathogenic potential, homologs were found in a range of host-associated microbes, which may be suggestive of cryptic pathogenic potential. Of the known pathogenic species, 11% are associated with human pathogenicity, including *Pseudomonas fluorescens, Photorhabdus asymbiotica, Serratia marcescens* and several species of *Providencia* (*P. alcalifaciens, P. rettgeri*, and *P. stuartii*); these bacteria are generally opportunistic pathogens, and are associated with severe diseases in immunocompromised individuals[38–43]. Interestingly, the majority of remaining LCT-T homolog sequences occurred in species associated with pathogenicity in non-human hosts, including species of *Vibrio,*

*Pseudomonas, Xenorhabdus,* and *Photorhabdus*, which are known pathogens of aquatic organisms, insects and fungi[44,45]. Notably, species of *Pseudomonas, Xenorhabdus* and *Photorhabdus* produce insecticidal toxins FitD and Mcf, which have previously noted homology to the TcdA/TcdB T-domain[46]. Although the association of LCT-T containing proteins with disease or infection is not known, their presence in pathogenic species strengthens the claim of a putative toxin functionality.

**LCT-T homologs contain diverse effector types**. We next annotated the individual domains within all 1104 LCT-T homologs, to determine the types of domains found within proteins containing LCT-like T-domains (and thereby identify potentially translocated effector domains). In support of a putative function as a toxin, most LCT-T homologs contain one or more LCT-like domain, with ~30% of sequences containing a glucosyltransferase (GTD-containing), ~20% with a glucosyltransferase and cysteine protease (GTD-CPD, or 'LCT-like') and ~10% with a cysteine protease (CPD-containing) (Fig. 2a). We also identified proteins containing different toxin domain families N-terminal to LCT-T domains (e.g., a homolog of anthrax toxin lethal factor in WP_102423241.1 from *Vibrio* sp. 10 N.261.52.A1). Interestingly, in ~40% of LCT-T containing proteins, the region N-terminal to the translocase-like domain is unannotated. These unannotated regions could be explained by the presence of known LCT-related domains that fall below detection thresholds, or else potentially represent uncharacterized domains. By comparison, the putative effector types associated with DT and BoNT translocation domains were predominantly ADP-ribosyltransferases (ADPR) and peptidases, respectively, the well-known effectors of DT and BoNT (Fig. 2b).

With respect to the size of the translocated cargo, we found that the LCT-Ts on average translocate much larger products: LCT effectors average 965 amino acids, compared to 479 amino acids for BoNT-T homologs and 218 amino acids DT-T homologs (Fig. 2b). Moreover, we found that the cargo potentially

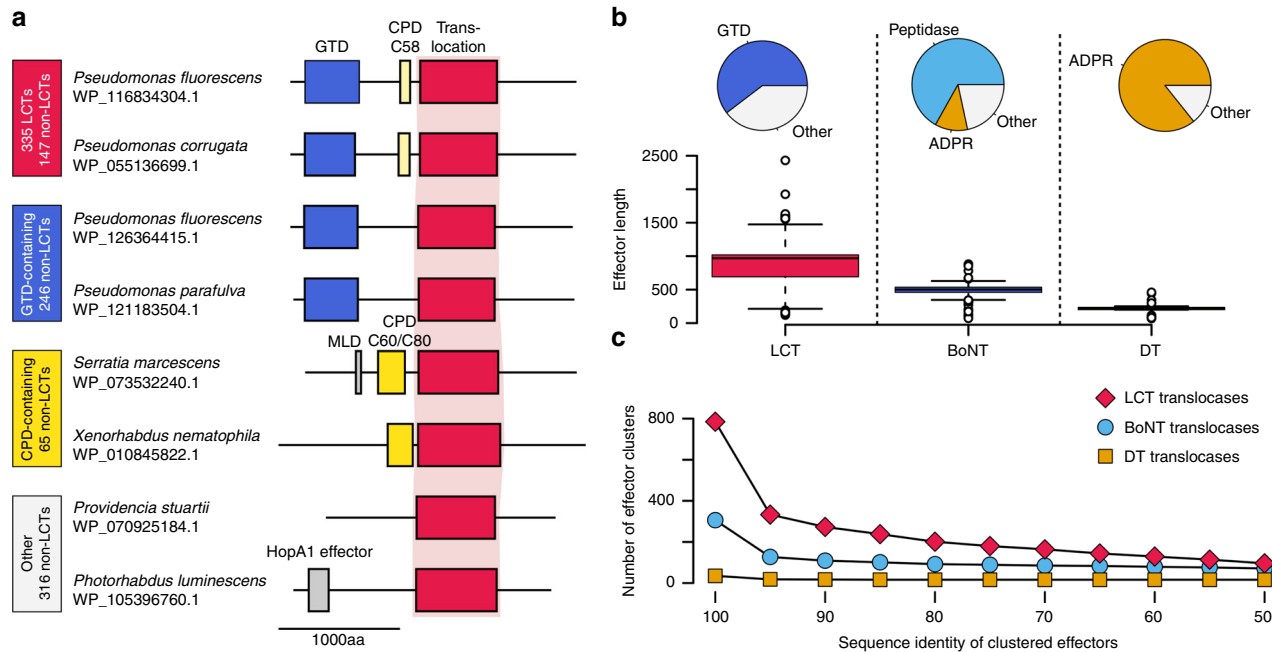

**Fig. 2 Analysis of proteins with LCT-like T-domains. a** Proteins with LCT-like T-domains can be found in three domain architecture types: LCT-like (red; 147 sequences), GTD-containing (blue; 246 sequences), CPD-containing (yellow; 65 sequences), and variable (other) composition (gray; 316 sequences). Within each domain architecture, two examples are shown. **b** By clustering the effectors at different levels of sequence identity, the diversity of effectors between the translocation domain families can be compared. At all levels of identity, the LCT family yields the largest number of effector sequence clusters, suggesting the LCT T-domain transports more diverse cargo than BoNT or DT translocases. **c** Comparison of effector lengths and types between BoNTs (blue), DTs (orange), and LCTs (red). The DT family possesses the smallest effectors, most of which are ADP-ribosyltransferases (ADPRs). The BoNT effector family is comprised mostly of metallopeptidases (peptidases) and ADPRs. The LCT family possesses the largest effectors, which are mostly GTDs.

translocated by LCT-T domains consistently possessed greater sequence diversity than those associated with BoNT or DT, across all levels of sequence identity (Fig. 2c). These data suggest that although each AB toxin-translocase may be fine-tuned for translocating particular types of effector, LCT-T translocases may be capable of translocating diverse effectors with a wider range of sequence and size diversity.

**S. marcescens protein is a functional translocating toxin.** To demonstrate that an LCT-T homolog could function as a trans-locating toxin, we characterized a distant homolog derived from *Serratia marcescens* (WP_073532240.1). Demonstrating even partial functionality for the *S. marcescens* (Sm) homolog, which lacks an annotated effector domain and shares only 20.2% sequence identity TcdB T-domain, would support the notion that the many remaining homologs that are more closely related to TcdB are also functional translocases (Fig. 2a). The Sm homolog is a 250-kDa protein containing an unannotated 63-kDa amino-terminal sequence that is upstream of a C80 peptidase, the LCT-T region and an unannotated carboxy-terminal region (Fig. 2a).

To characterize Sm function, we initially evaluated the ability of the individual domains to function in a manner consistent with a functional toxin. First, we transfected human HeLa cells with two versions of the putative Sm toxin effector: (Sm 1–600), containing the entire sequence upstream of the suspected domain boundary with CPD; and, (Sm 36–600) in which the amino-terminal 35 largely hydrophobic residues were removed. In both cases, intracellular expression of Sm putative effector sequences resulted in profound cell-rounding compared to the empty plasmid control (Fig. 3a). Next, to confirm the autoprocessing functionality of the C80 peptidase, we incubated full-length Sm with the allosteric activator inositol hexakisphosphate (InsP6) in vitro. In the presence of Insp6, Sm undergoes autoproteolysis,

yielding two predominant fragments on SDS-PAGE that are consistent with release of a ~60–65-kDa fragment from the full-length toxin (Fig. 3b). The pH-dependent pore-forming ability of Sm was next evaluated by measuring dye release from liposomes that were pre-loaded with the quenched dye-pair HPTS/DPX. Whereas at neutral pH, no dye release from liposomes was seen in the absence or presence of Sm, we see significant dye release at low pH in the presence of Sm (Fig. 3c, d). Lastly, we evaluated whether Sm could induce morphological effects when added to cells. Though the particular host cell and host cell receptor(s) for Sm are not known, we nevertheless observed dose-dependent cell-rounding of human colorectal cells (HCT-116) by Sm albeit at higher doses that are generally for the human-specific LCT family toxins (Fig. 3e).

**LCT-T homologs have an evolutionarily conserved translocase.** Next, we leveraged the greatly expanded number and diversity of identified LCT-T homologs to uncover conserved molecular features of the T-domain. Alignment of the TcdB T-domain and LCT-T homologs revealed a shared core region, with a distribution of start and end sites at amino acids 815 (±6 residues) and 1514 (±99 residues), respectively (Fig. 4a); hereafter, we will refer to this region as the evolutionarily conserved translocase (ECT). In the context of the best characterized homolog, TcdB, the ECT encompasses regions previously implicated in pore-formation and translocation[21,22]. Within the ECT there are three distinct regions (region i: 956–1019; region ii: 1029–1078; region iii: 1090–1110) that share a remarkably similar pattern of hydropathy—one small peak, followed by two larger peaks—that map to putative membrane-insertion regions[21] (Fig. 4b). Furthermore, many residues that are conserved within region ii and region iii among the LCT-T homologs (TcdB residues: I1035, D1037, L1041, P1095, G1098, I1099, L1106, and V1107) correspond to

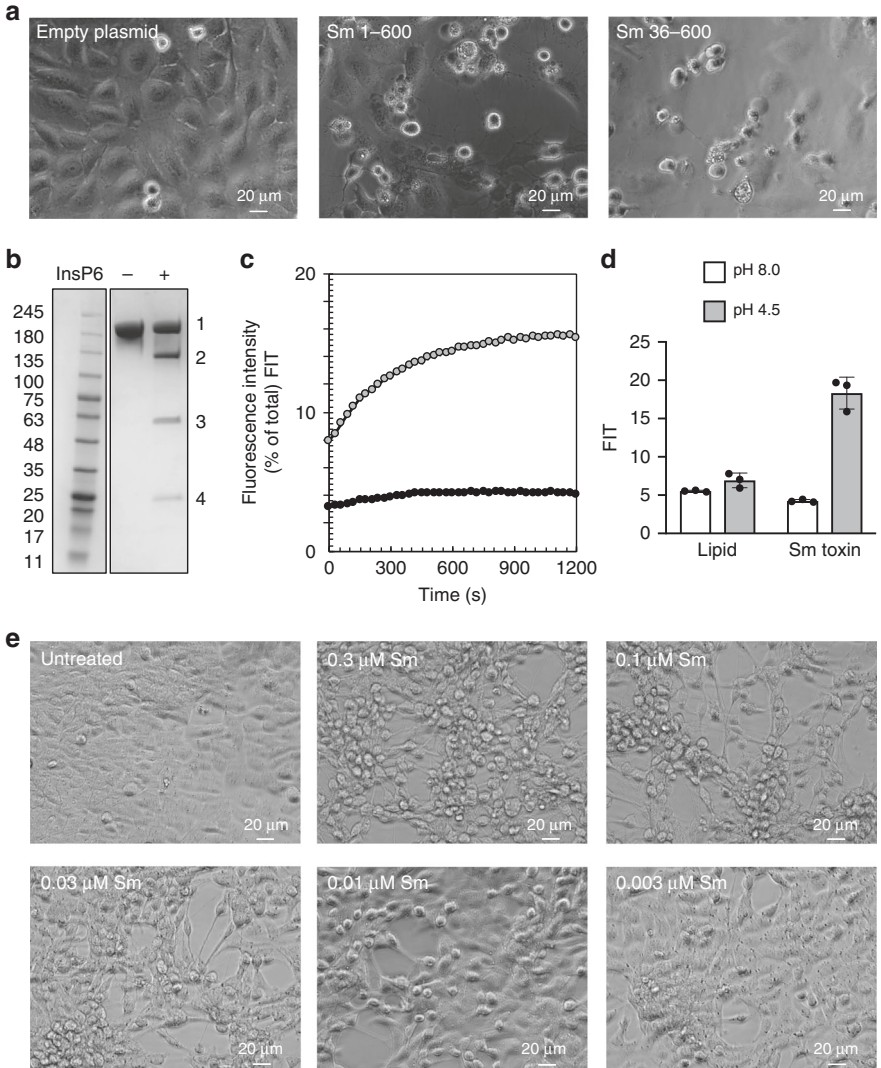

**Fig. 3 Characterization of *S. marcescens* (Sm) toxin. a** Light microscopy images of HeLa cells transfected with PiggyBac Dual Promoter PB5131B-1 vector, expressing empty plasmid or Sm toxin effector (1–600 or 36–600) after 24 h. **b** Addition of Insp6 (+) to Sm toxin results in cleavage of the effector (band 3) and the CPD (band 4) from the full-length toxin (band 1). Band 2 represents full-length Sm toxin without the effector and CPD. **c** Dye release from HPTS/DPX loaded liposomes at pH 4.5 and pH 8.0. **d** Quantification of dye release from HPTS/DPX loaded liposomes after 20 minutes. $N = 3$ for all experiments. Source data are provided as a Source Data File for Fig. 3e. **e** Light microscopy images of HCT-116 cells with no toxin (untreated) or with varying concentrations of Sm toxin after 24 h.

residues in TcdB that are implicated in pore formation and/or translocation[22] (Fig. 4c). In addition to important single residues, it is intriguing that, similar to BoNT and DT[29], a PxxG (more specifically, PxxGL) motif was identified as being strongly conserved in LCT-T homologs.

Identification of an evolutionarily-conserved translocase (ECT) in LCT-T homologs led us to hypothesize that such smaller-sized forms of the larger LCT T-domain might comprise the core machinery that is necessary and sufficient for pore-formation and translocation. To address this directly, we used the relative distribution frequency of start and end sequence coverage sites as guides to design a series of T-domain truncations in the most well-characterized LCT homolog, TcdB, generating TcdB truncations with two different N-terminal start sites (*viz.*, residues 800 and 851), and a variable C-terminus ($X = 1500, 1473, 1394, 1338$) (Fig. 5, Supplementary Fig. 6a). In order to assess translocation in cell-based assays, we developed a pore-formation/translocation platform using the DT ADP-ribosyltransferase (ADPR) and the DT receptor-binding domain (DTR) as a scaffold, such that

test chimeras would have the general ADPR-[truncated TcdB T-domain]-DTR architecture. We used DT because of the well-established and facile readout of the ADPR (protein synthesis) and the robust binding of DTR to the ubiquitous HBEGF receptor, present on many cell lines. Practically, the ADPR and DTR domains are amenable to greater levels of expression in *E. coli* over the GTD, CPD of LCTs, making a DT-TcdB chimera a more feasible platform to screen a large number of constructs.

We subjected the chimeras to two rounds of experimental testing, first testing their ability to form pores, and then, their ability to translocate. To probe pore formation, we measured dye release from pre-loaded liposomes, and to assess translocation, we evaluated intoxication (indirect measure of ADPR translocation into the cytosol) and protein synthesis inhibition (direct measure of ADPR translocation into the cytosol). All TcdB truncations formed pores in our dye release assay (Fig. 5b, c); however, only two truncations were able to facilitate translocation (i.e. 800–1500, 800–1473) (Fig. 5d, Supplementary Fig. 7); constructs were considered non-toxic if unable to intoxicate cells at concentrations

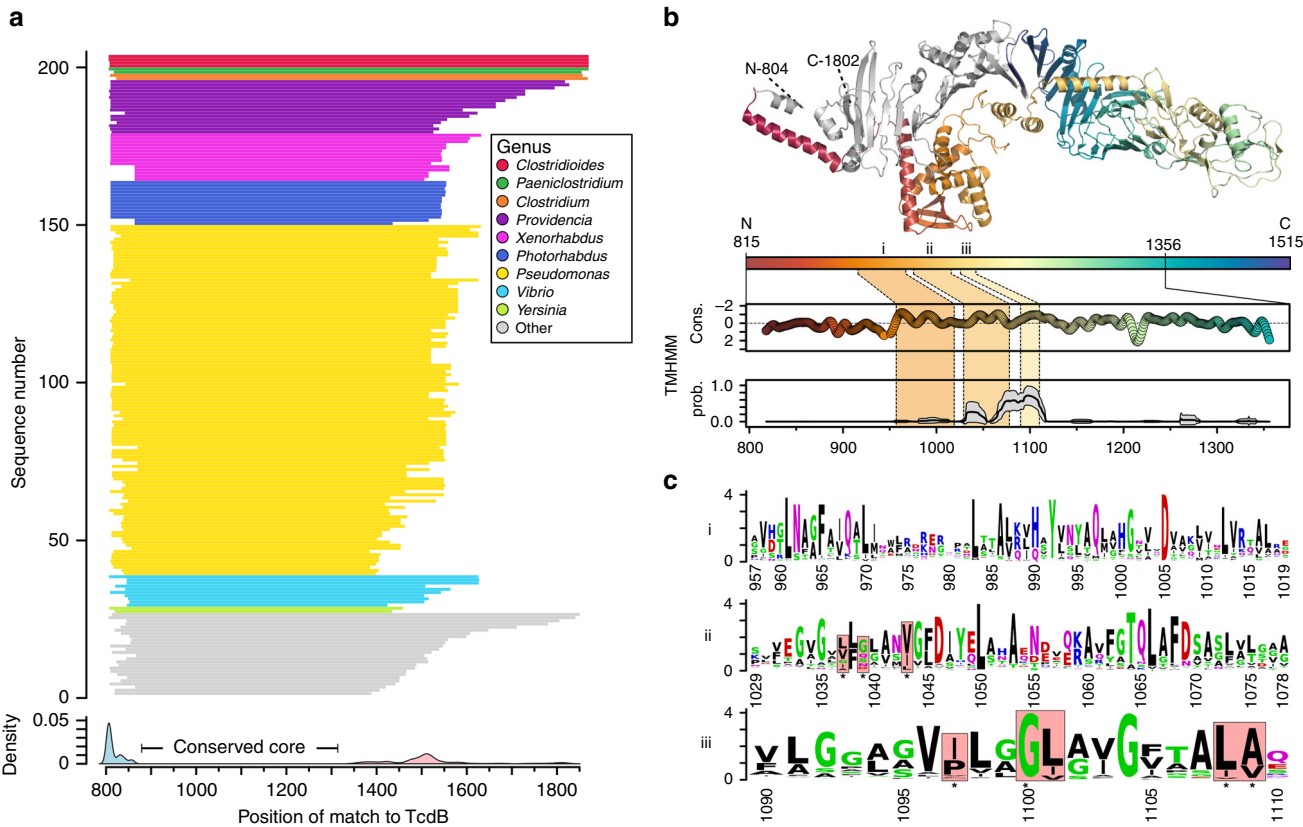

**Fig. 4 Patterns in sequence conservation among LCT-like T-domains. a** The distribution of BLAST matches to the TcdB T-domain demonstrates that the region common to all LCT-like T-domains is limited to within the first 700 amino acids of the N terminus. **b** The structural context of the mean BLAST start (815) and end (1515) sites, as mapped on to the TcdA T-domain (PDB 4R04). Normalized sequence conservation for the region spanning residues 815 to 1356 shows three main peaks of sequence conservation (i, ii, and iii). Two of these conserved regions correspond to an average increase in predicted transmembrane helix propensity. **c** Within the three sequence conservation peaks, several sites known to result in the loss of translocation activity in TcdB are conserved among LCT-like T-domains. However, several strongly conserved residues are not associated with a loss of translocation function, suggesting they are related to some other key function in LCT-like T-domains.

up to 50 nM (Supplementary Fig. 7a). In line with the above data, we found that all TcdB T-domain truncations starting at residue 851 formed pores (Fig. 5f, g), while only 851–1500, 851–1473 were able to facilitate translocation (Fig. 5h, Supplementary Fig. 7). Further truncation to 881 (i.e. 881–1473) abrogated translocation (Supplementary Fig. 7a). Taken together, these results indicate that residues 851–1473 comprise all of the components needed for pore-formation and translocation.

To further interrogate the evolutionarily conserved (and minimal TcdB) translocase, we also produced TcdB 851–1473 in a hybrid TcdB-DT system, with the GTD and CPD of TcdB and the receptor-binding domain of DT, such that the chimera was GTD-CPD-[TcdB(851–1473)]-DTR (Fig. 6a, Supplementary Fig. 6b). To evaluate translocation, we assessed cell rounding and Rac1 glucosylation (both direct measures of GTD translocation into the cytosol) and intoxication (indirect measure of GTD translocation into the cytosol). TcdB 851–1473 caused cells to round (Fig. 6b), glucosylated Rac1 (Fig. 6c) and intoxicated cells (Fig. 6d), while GTD-CPD-DTR lacking any of the TcdB T-domain did not cause cells to round and was non-toxic, reinforcing TcdB 851-1473 as an evolutionarily conserved and functional translocase region.

**The ECT is an autonomous pH-dependent translocase**. In context of the entire TcdB T-domain, the ECT extends from one end of the T-domain to the other, is a mixture of both helical and β-sheet content and generally does not look like an independently folded protein domain (Fig. 7a); further, no studies have shown that smaller fragments of the T-domain retain pore formation and translocation activity. We recombinantly produced the ECT from TcdB, which is strikingly soluble, stable and amenable to characterization (Supplementary Fig. 6c). The ECT on its own was still functionally active, exhibiting pH-dependent pore formation, with maximal dye release at pH 4.0 and minimal dye release above pH 5.0 (Fig. 7b). At low pH and in aqueous solution (i.e. pH 4.0, 4.5, 5.0), the ECT rapidly aggregated out of solution as expected. By contast, in presence of the membrane mimetic, dodecylphosphocholine (DPC) the ECT remained in solution and soluble (Fig. 7c, d). Interestingly, in contrast to its structure in context of the full-length T-domain, the ECT had characteristic circular dichroism (CD) spectra of a helical protein, suggesting the ECT undergoes structural changes en route to and when inserted into the membrane (Fig. 7e, f). Taken together, our data indicate that the ECT is an autonomous, folded and functionally active protein translocase.

## Discussion

In this work, we conducted a targeted search to identify proteins that have homology to the T-domain of TcdB—the best characterized member of the small LCT family, of which there was previously only 6 total members (TcdA, TcdB, TcsL, TcsH, TpeL, and TcnA). Querying just the T-domain of TcdB, rather than the entire toxin, enabled identification of LCT-T homologs in bacteria outside of clostridia, and outside the conventional LCT

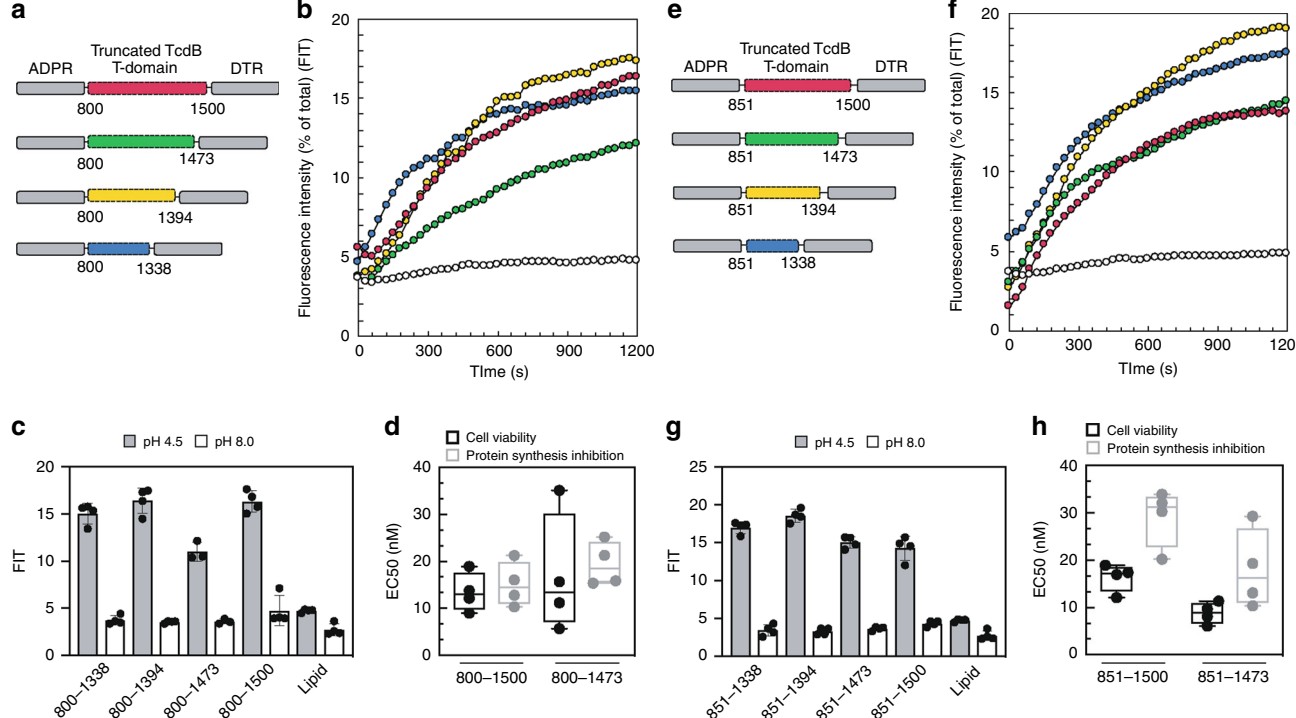

**Fig. 5 Defining the evolutionarily conserved and minimal LCT translocase region. a** Construct design of ADPR-[truncated TcdB T-domain]-DTR chimeras. **b** Dye release from HPTS/DPX loaded liposomes at pH 4.5, with truncated TcdB T-domains colored as follows: 800–1500 (red), 800–1473 (green), 800–1394 (yellow), and 800–1338 (blue). **c** Quantification of dye release from HPTS/DPX loaded liposomes after 20 min, at pH 4.5 and pH 8.0 ($N = 3$). **d** Box plot of $EC_{50}$ values from cell viability and protein synthesis inhibition of Vero cells ($N = 4$), where center line is the median, bounds of box represent the interquartile range, and the wiskers represent the minimum and maximum of data. Panels **e–h** the same as **a–d**, respectively, but with truncated TcdB T-domains 851–1500 (red), 851–1473 (green), 851–1394 (yellow), and 851–1338 (blue). Source data are provided as a Source Data File for Fig. 5**c**, **g**, **d**, **h**.

domain architecture (i.e., also containing a GTD, CPD, and CROPs). Despite different genomic contexts and diverse upstream effectors, LCT-T homologs have—and are linked—by an ECT.

The ECT has important implications for our understanding of LCT translocation. Primarily, the ECT is a functional translocation unit that exists within the larger scaffold of the T-domain. Within the LCT T-domain, the ECT is entwined with receptor binding moieties, which is an unexpected positioning of a functional domain. Although it is not yet clear how the ECT functions in context of the rest of the T-domain at low pH or a membrane, it is evident that the ECT does not require the rest of the T-domain to facilitate translocation. The ECT therefore reflects the minimal necessary and sufficient requirements for translocation.

Within the ECT, LCT-T homologs clarify important translocation features. Principally, important residues for LCT translocation are strongly conserved in distant LCT-T homologs, indicating an essential role of these residues in translocation, that seems to persist independent of the organism and the translocating effector. Analysis of LCT-T homologs also reveals strongly conserved residues—and therefore, potentially important translocation residues—that are not apparent when only comparing sequences of the LCT family. Importantly, the conservation of residues and hydropathy pattern within the ECT suggests that LCTs and LCT-T homologs have highly similar membrane-inserted structures and translocation mechanisms, and that homologs could be used in mechanistic studies of translocation.

On the basis of the overall architectures of the hundreds of identified LCT-T homologs, many of which contain both a known cytotoxic effector domain and a toxin-derived autoprocessing domain that are upstream of an ECT that is replete with key sequence motifs that are essential for pore-formation and translocation[22], we contend that the large majority of the proteins described here have the capacity to function as toxins. To demonstrate this, in part, we selected a distant LCT-T homolog from *S. marcescens* with an unannotated effector and low sequence identity within the ECT and showed that it was capable of inducing cell rounding by intracellular delivery of its effector. Although our work supports the claim of Sm as a toxin, it is important to note that we do not know the relevance of Sm toxin to *S. marcescens* infections. As is the challenge for many bioinformatically identified toxins, clarifying the toxin role in virulence is key to understanding toxin functionality. Clarifying the role of Sm toxin—and other LCT-T homologs—requires identifying a relevant host, and fulfillment of Falkow's molecular Koch's postulates[47], which require that toxins (or virulence factors) exist only in a pathogenic strain, with mutation or deletion of the virulence factor resulting in loss of pathogenicity. We hope our work provides a starting point and framework to further interrogate the function and ecological significance of these hundreds of putative toxins.

## Methods

**Detection of LCT-T homologs and dataset curation**. The TcdB T-domain (UniProt ID P18177.3, residues 800–1814) was used as a query for two iterations of PSI-BLAST[48] (with default parameters: BLOSUM62 substitution matrix, gap existence 11, gap extension 1) against the NCBI non-redundant protein database (nr) on 13 June 2019. A total of 1573 protein sequences were retrieved, 1216 of which yield $E$-values less than 1e-5 after a single BLASTP search. Proteins labeled as 'partial' or otherwise truncated, as well as any proteins with <100 amino acids upstream of the translocase were removed from the dataset, leaving a final set of 1104 translocase sequences. In order to verify the relationship of the query to these sequences, pairwise comparisons between TcdB to target translocases shuffled 10,000 times were performed using the FASTA3 package (v. 36.3.8), 1023 of which produce $E$-values less than 1e-5.

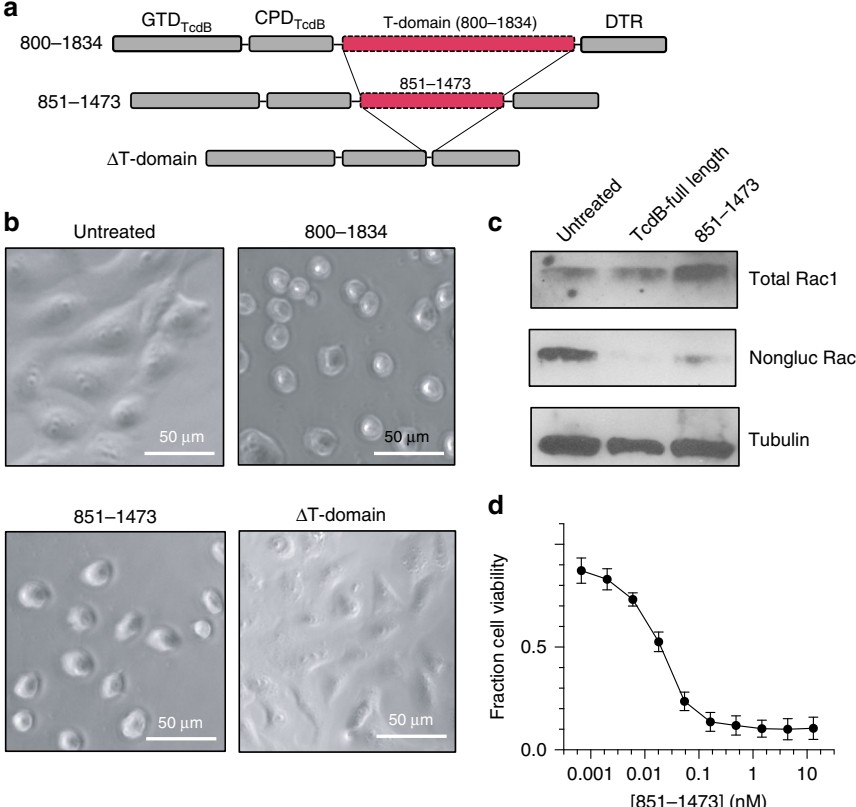

**Fig. 6 TcdB 851-1473 functions as a translocase in a hybrid TcdB-DT context. a** Construct design of GTD-CPD-[TcdB(800–1834)]-DTR (800–1834), GTD-CPD-[TcdB(851–1473)]-DTR (851–1473) and GTD-CPD-DTR (ΔT-domain). **b** Light microscopy images of Vero cells with no toxin (untreated), 800–1834, 851–1473 or ΔT-domain after 3 h. All proteins were added to cells at a final concentration of 1 nM. **c** Western blot to detect total Rac1, non-glucosylated Rac1 in untreated Vero cells, and in cells treated with TcdB or 851–1473. All proteins were added to cells at a final concentration of 1 nM. **d** Fraction cell viability of Vero cells incubated with 851–1473, with 1.0 indicating 100% viable cells and 0 indicating 0% viable cells. $N = 4$ for all experiments. Source data are provided as a Source Data File for Fig. 6c.

**LCT-T homologs correlating domains and metagenomic abundance.** AnnoTree was used for phylogenomic visualization of species containing LCT-T homologs across the bacterial tree of life[36]. Proteome-level Pfam annotations from the AnnoTree database were used to determine the domains most correlated with an annotated LCT pore-forming domain model (Pfam model PF12920), the top 100 of which are available in Supplementary Table 1. Metagenomic surveys were performed using EBI's MGnify server (http://www.ebi.ac.uk/metagenomics) using the TcdB translocase (UniProt ID P18177.3, residues 800–1814) as the query.

**LCT-T homologs associated pathogenicity.** The association of bacterial organisms with pathogenicity was estimated based on where the organism was isolated and reviewing the literature, where possible. Broadly, an organism's level of pathogenicity was categorized into one of four possibilities: no known pathogenicity or host association, host-associated with no known pathogenicity, known pathogen of non-human hosts, and known pathogen of humans. A detailed spreadsheet can be found in the source data file (Supplementary Fig. 5).

**Comparison of effector diversity from AB toxin families.** The effector domains from different toxin families were retrieved by searching with each toxin's translocase domain as a query (BoNT: PDB identifier 3BTA, residues 548–865; DT: SwissProt identifier P00588.2, residues 232–383) against the NCBI non-redundant protein database with two iterations of PSI-BLAST. For DT and BoNT, the entire portion N-terminal to the translocase hit region was extracted and treated as the effector region. For the LCT family, the effector region was more difficult to define because it contains the glucosyltransferase domain as well as the autoproteolytic cysteine peptidase domain, and not all LCT-T homologs have detectable peptidase domains. Thus, the entire region N-terminal to the translocases in proteins lacking a peptidase, and the regions N-terminal to peptidases in peptidase-containing sequences, were extracted separately to yield the set of LCT effectors. The putative effector regions from BoNT, DT, and the LCTs were clustered at increments of 5% cluster sequence identity between 50 and 100% using USEARCH[49]. Effector types were assigned using InterProScan (v5.33-72)[50–54].

**Evolutionary analysis of the LCT-T domain.** The curated set of translocase sequences was further reduced to a set of 203 non-redundant sequences by clustering with USEARCH[49] (v10.0.240) at 90% identity. These sequences were aligned with the L-INS-I algorithm of the MAFFT package (v7.407)[55], and a maximum likelihood tree was inferred using RAxML (v8.2.4)[56] with automatic evolutionary model selection (LG), 4 gamma-distributed rate categories, automatic bootstrapping with autoMRE, and a thorough ML search. Protein domains were annotated using InterProScan (v5.33-72)[50–54]. AnnoTree was used for phylogenomic visualization of species containing LCT-T homologs across the bacterial tree of life[36].

Conservation scores were calculated for residues in the reduced set of 203 translocase sequences using the ConSurf web server[57]. The structural context of these sites was depicted on the TcdA structure (PDB 4R04) using PyMol (https://pymol.org). Transmembrane helix prediction was estimated using the TMHMM2.0 server[58]. Motifs were depicted using WebLogo[59].

**Generation of recombinant protein.** Full-length Sm protein (WP_073532240.1) lacking the first 35 amino acids was synthesized and codon-optimized for expression in *E. coli* (GenScript) and fused into a pET28a vector using In-Fusion HD cloning (Clontech). Sm toxin residues 1–35 were removed to improve solubility during purification. The source data file (tab Sm toxin) contains the Pfam annotations and hydropathy plot for Sm toxin. Regions of the TcdB T-domain were amplified from a codon-optimized TcdB gene for expression in *E. coli* (GenScript) and fused into a pET28a vector using In-Fusion HD Cloning (Clontech). For ADPR-[truncated TcdB T-domain]-DTR chimeras, TcdB T-domain was fused into a vector containing the diphtheria toxin ADP-ribosyltransferase (ADPR) (defined here as residues 1–201) with an intact furin cleavage site and diphtheria toxin receptor binding region (DTR) (defined as in DT as amino acids 378–535). For GTD-CPD-[truncated TcdB T-domain]-DTR chimeras, the TcdB T-domain was fused into a vector contained the TcdB GTD and CPD (defined here in TcdB as residues 1–543 and 544–799, respectively) and DTR. For regions of the TcdB T-domain truncation beginning at 851 or 881, a short linker (four glycine followed by one serine ($G_4S$)) was added between the truncated TcdB T-domain, and the CPD or ADPR.

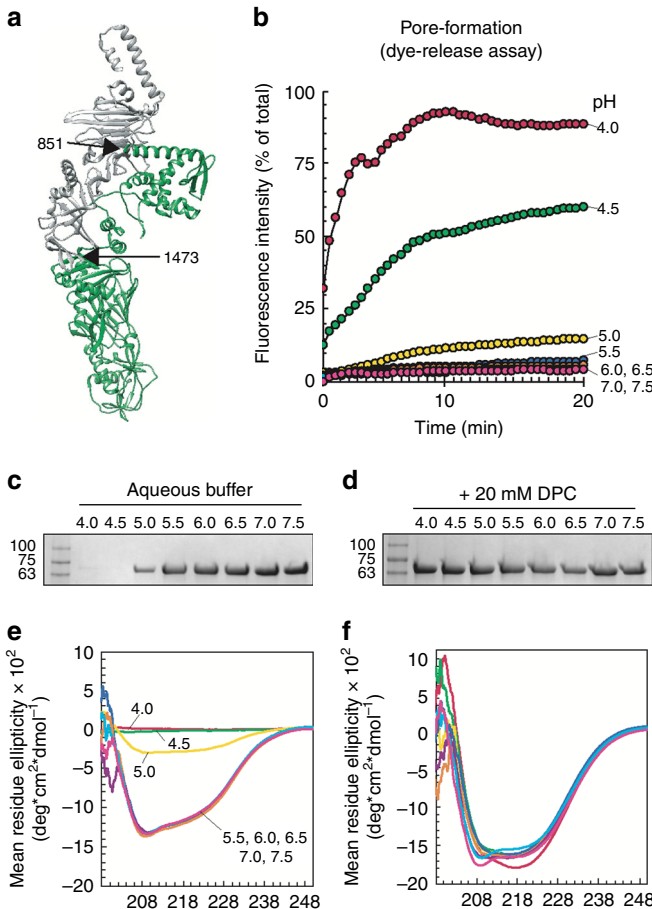

**a** TcdA T-domain

**b** Pore-formation (dye-release assay)

**c** Aqueous buffer

**d** + 20 mM DPC

**e**

**f**

**Fig. 7 TcdB 851-1473 is an autonomous protein domain. a** TcdA T-domain (pdb: 4R04), with 851–1473 colored in green. **b** TcdB 851–1473 induced dye release from HPTS/DPX loaded liposomes from pH 4.0 to pH 7.0 in 0.5 pH increments. **c** Stability of TcdB 851–1473 in aqueous buffer and with **d** 20 mM DPC from pH 4.0 to pH 7.5 in 0.5 pH increments. **e** Circular dichroism (CD) spectroscopy of TcdB 851–1473 in aqueous buffer and **f** 20 mM DPC from pH 4.0 to pH 7.5 in 0.5 pH. Coloring for each pH in **f** is the same as in **e**. All experiments are $N = 3$. Source data are provided as a Source Data File for Fig. 7c, d.

**Expression and purification of recombinant protein**. Sm toxin, toxin chimeras and TcdB 851–1473 were transformed into *E. coli* BL21 DE3 competent cells and expressed with N-terminal $H_6$-SUMO and C-terminal Strep tags (Sm toxin, toxin chimeras) or with an N-terminal $H_6$-SUMO tag (TcdB 851–1473). For all proteins, a total of 20 mL of overnight culture was inoculated into 1.0 L of TB with 50 μg/mL kanamycin and induced at $OD_{600}$ ~ 0.6–0.8 with 1 mM isopropyl-β-thiogalactopyranoside (IPTG) at 25 °C (toxin chimeras) for 4 h or 18 °C overnight (Sm toxin and TcdB 851–1473). Cells were harvested by centrifugation and resuspended with lysis buffer (20 mM Tris pH 8.0, 500 mM NaCl) and lysed by an EmusliFlex C3 microfluidizer (Avestin) at 15,000 psi. Whole cell lysates were then centrifuged at 15,000 × *g* for 20 min and the supernatants were passed through a 0.2-μm filter. Toxin chimeras were purified by Strep-Tactin affinity chromatography using a Strep-Tactin column (GE Healthcare) and eluted in 20 mM Tris pH 8.0, 150 mM NaCl, 1 mM D-desthiobiotin and 5% glycerol. Sm toxin and TcdB 851–1473 were purified by nickel affinity chromatography using a Ni-Penta column (Marvelgent biosciences) and eluted with an imidazole gradient. The $H_6$-SUMO tag of Sm toxin and TcdB 851–1473 was removed by adding 1U of Sumo protease (Life Sensor), 1 mM DTT (Thermo Scientific) and incubating at 25 °C for 2 h. Sm toxin and TcdB 851–1473 were dialyzed to 20 mM Tris pH 8.0, 500 mM NaCl to remove imidazole and purified using His-Pure Ni-NTA resin (Thermo Scientific) to remove the $H_6$-Sumo protease and $H_6$-Sumo tag from the purified protein samples. Sm toxin was further purified by Strep-Tactin affinity chromatography as detailed above. All proteins were verified by SDS-PAGE, concentrated with a 30,000 MWCO ultracentrifugation device. Protein concentration was calculated by densitometry using ImageJ software.

**In vitro autoprocessing assay**. In all, 5 μg of Sm toxin in 50 mM Tris pH 8.0 was incubated with 5 mM DTT ±500 μM Insp6 (Thermo Scientific) for 20 min at 37 °C before stopping with Laemmli loading buffer with beta-mercaptoethanol (Bio-Rad). Insp6 induced autoprocessing was assessed by electrophoresing the samples on SDS polyacrylamide gels and staining with Coomassie Blue R250.

**Cell rounding by Sm toxin**. HCT-116 cells (ATCC, Cat #CCL-247) were cultured in McCoy's 5 A medium (Wisent) with 10% FBS (Wisent) and 1% penicillin/streptomycin (Wisent). HCT-116 cells were seeded at a density of 8000 cells per well in 96-well plates (Corning) and cultivated at 37 °C and 5% $CO_2$ overnight. The next day, Sm toxin was added to HCT-116 cells in a serial dilution of 1/3 and incubated at 37 °C and 5% $CO_2$. After 24 h, light microscopy images were taken to assess cell morphology.

**Transfection of Sm toxin effector**. HeLa cells (ATCC, Cat #CCL2) were cultured in DMEM (Wisent) with 10% FBS (Wisent) and 1% penicillin/streptomycin (Wisent). HeLa cells were seeded at a density of 8000 cells per well in 96-well plates (Corning) and cultivated at 37 °C and 5% $CO_2$ overnight. The next day, PiggyBac Dual Promoter PB5131B-1 vector (System Biosciences) with the effector region of Sm toxin co-expressing GFP was transfected into HeLa cells using FuGENE HD Transfection Reagent (Promega). To assess changes in cell morphology from the transfection reagent and the plasmid, an empty PiggyBac vector lacking the Sm effector was transfected as a control. Transfection efficiency was assessed after 24 h by GFP fluorescence, and light microscopy images were taken to assess the morphology of transfected cells.

**HPTS/DPX dye release from liposomes**. The HPTS/DPX dye release assay was based on protocols from Genisyuerek et. al.[23]. In brief, liposomes were prepared with 1,2-dioleoyl-sn-glycero-3-phosphocholine (DOPC) (Avanti Polar Lipids), with 0.8% 1,2-dioleoyl-sn-glycero-3-[(N-(5-amino-1-carboxypentyl)iminodiacetic acid)succinyl] (nickel salt) (DGS-NTA[Ni]) (Avanti Polar Lipids). After drying down with $N_2$, the lipid film was resuspended in 20 mM Tris, 150 mM NaCl, pH 8.0, 35 mM 8-Hydroxypyrene-1,3,6-trisulfonic acid (HPTS) and 50 mM p-xylene-bis-pyridinium bromide (DPX) (Thermo Fischer). Lipid vesicles were subjected to 10x freeze-thaw cycles and extruded using a 200 μm filter. To get rid of un-encapsulated dye, the lipid vesicles were then subjected to gel filtration and eluted in 20 mM Tris, 150 mM NaCl. To assess fluorophore leakage, protein were added in a ratio of 1:10,000 with liposomes, such that the final liposome concentration was ~400 μM. The fluorescence was monitored in a 96-well opaque plate (Corning) (excitation 403 nm, emission 510 nm) in high pH buffer (20 mM Tris, 150 mM NaCl, pH 8.0) or low pH buffer (20 mM Na-acetate, 150 mM NaCl, pH 4.5), or with citrate-phosphate buffers ranging from pH 4.0-pH 7.5 in 0.5 pH increments. To determine total HPTS fluorescence, Triton X-100 was added to each well to a final concentration of 0.3%. All spectra were normalized to 100% dye release by 0.3% triton. Quantification of dye release data (Figs. 3d, 5c, g) is provided in the source data file.

**Cell viability**. Vero cells (ATCC, Cat #CCL-81) were cultured in DMEM (Wisent) with 10% FBS (Wisent) and 1% penicillin/streptomycin (Wisent). Vero cells were seeded at a density of 4000 cells per well in 96-well plates (Corning) and cultivated at 37 °C and 5% $CO_2$ overnight. Toxin chimeras were added to Vero cells in a serial dilution of 1/3 and incubated at 37 °C and 5% $CO_2$. After 48 h, cell viability was assessed by PrestoBlue Cell Viability Reagent (Life Technologies). Fluorescence was read on a Spectramax M5 plate reader (Molecular devices). For each toxin condition, the data were blank subtracted (i.e. buffer only, no cells) and normalized (cells, untreated, representing 100% viable cells) and converted to fraction viable cells. From these data, dose response curves and half-maximal effective concentrations ($EC_{50}$ values) were generated using Prism software. Individual $EC_{50}$ values (Fig. 5d, h, Supplementary Fig. 7a) are provided in the source data file.

**Protein synthesis inhibition**. Vero cells (ATCC, Cat #CCL-81) stably expressing NanoLuc (Nluc) Luciferase (Promega) were cultured in DMEM (Wisent) with 10% FBS (Wisent) and 1% penicillin/ streptomycin (Wisent). Vero Nluc cells were seeded at a density of 4000 cells per well in 96-well plates (Corning) and cultivated at 37 °C and 5% $CO_2$ overnight. Toxin chimeras were added to Vero NLuc cells in a serial dilution of 1/3 and incubated at 37 °C and 5% $CO_2$ for 24 h. Nano-Glo Luciferase Assay substrate and buffer (Promega) were added to cells as per the manufacturer's instructions, and luminescence was read on a Spectramax M5 plate reader at (Molecular Devices). For each toxin condition, the data were blank subtracted (i.e. buffer only, no cells) and normalized (cells, untreated, representing 0% protein synthesis inhibition) and converted to fraction protein synthesis inhibition. From these data, dose response curves and half-maximal effective concentrations ($EC_{50}$ values) were generated using Prism software. Individual $EC_{50}$ values (Fig. 5d, h) are provided in the source data file.

**Cell rounding by toxin chimeras**. Vero cells (ATCC, Cat #CCL-81) were cultured in DMEM (Wisent) with 10% FBS (Wisent) and 1% penicillin/ streptomycin

(Wisent). Vero cells were seeded at a density of 8000 cells per well in 96-well plates (Corning) and cultivated at 37 °C and 5% $CO_2$ overnight. The next day, media was exchanged with serum-free media and cells were intoxicated by adding toxin chimeras at 1 nM. After 3 h, light microscope images were taken to assess rounding of cells.

**Rac1 glucosylation.** Vero cells (ATCC, Cat #CCL-81) were cultured in DMEM (Wisent) with 10% FBS (Wisent) and 1% penicillin/ streptomycin (Wisent). Vero cells were seeded at a density of 100,000 cells per well in 6-well plates (Corning) and cultivated at 37 °C and 5% $CO_2$ overnight. The next day, media was exchanged with serum-free media and cells were intoxicated by adding toxin at 1 nM. After 1 hr, media was aspirated from cells, cells were washed with PBS and lysed by addition of Laemmli loading buffer with beta-mercaptoethanol (Bio-Rad) to each well. Samples were heated to 90 °C before immediately loading on an SDS-PAGE gel. Following electrophoresis, samples were transferred to nitrocellulose using standard wet transfer protocols, blocked with 5% milk/ Tris-buffered saline (TBS) and probed for total Rac1 (1:1000 dilution) with Anti-Rac1 antibody 23A8 (Millipore Sigma, Cat #05-389) or for non-glucosylated Rac1 (1:1000 dilution) with Anti-Rac1 Mab102 (BD Biosciences, Cat #610651). Anti-α-tubulin (1:5000 dilution) (Sigma, Cat #T5168) was used as the loading control. Following overnight incubation with the primary antibody, the blot was washed with TBS/0.1% Tween20 and incubated with (1:10,000 dilution) with Anti-mouse conjugated horseradish peroxidase (GE Healthcare, Cat #NXA931V) for 60 min. After the final washes in Tris-buffered saline with Tween20, chemiluminescent detection was carried out using Clarity Western ECL Substrate (Bio-Rad) and exposing to Bio-Max MR film (Kodak). The uncropped Western blots (Fig. 6c) are in the data source file.

**Stability studies.** TcdB 851–1473 (5–10 μM) was incubated in citrate-phosphate buffers ranging from pH 4.0-pH 7.5 in 0.5 pH increments at room temperature in the presence and absence of 20 mM dodecylphosphocholine (DPC). After 30 minutes, samples were spun at $5000 \times g$ for 5 min to pellet aggregates (but not detergent). The supernatant was removed from each sample, and mixed 1:1 with Laemmli loading buffer with beta-mercaptoethanol (Bio-Rad) and boiled for 2 min. Samples were then loaded onto an SDS-PAGE gel and stained with Coomassie Blue R250. The uncropped SDS-PAGE gels (Fig. 7a) are in the data source file.

**Circular dichroism spectroscopy.** Far-UV CD spectra were recorded at room temperature using a J-810 spectropolarimeter (Jasco) with 0.1 cm path length cuvettes. Protein was added to a final concentration of 5–10 μM in presence or absence of 20 mM dodecylphosphocholine (DPC) in citrate-phosphate buffers ranging from pH 4.0–7.5 in 0.5 pH increments. After 30 min at room temperature, all samples were spun down at $5000 \times g$ for 5 min to pellet aggregates but not detergent. The supernatant was removed for each sample, and CD spectra were acquired from 250 to 190 nm at 50 nm/min, with a data pitch of 0.1 nm and three accumulations. Spectra were then averaged, blank subtracted and converted to mean residue ellipticity using standard formulas.

**Reporting summary.** Further information on research design is available in the Nature Research Reporting Summary linked to this article.

## Data availability
Data underlying Figs. 3d, 5c, g, d, h, 6c, and 7c, d, Supplementary Figs. 5 and 7a are provided as Source Data files. All other datasets generated during and/or analyzed during the current study are available from the corresponding authors on reasonable request.

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

## Acknowledgements
This work was supported in part by a Natural Sciences and Engineering Research Council of Canada Discovery Grant RGPIN-2017-06817 (to R.A.M.) and a CIHR Project Grant (to R.A.M.) and a Natural Sciences and Engineering Research Council Scholarship of Canada Graduate Scholarship-Doctoral (to K.E.O.). A.C.D. acknowledges funding from the Natural Sciences and Engineering Research Council of Canada (NSERC Discovery Grant RGPIN-2019-04266) and an Ontario Early Researcher Award.

## Author contributions
M.J.M. and A.C.D. performed the bioinformatics analysis. K.E.O conducted the experiments. K.E.O., M.J.M., A.C.D. and R.A.M. contributed to project design, analyzed data, and made contributions to writing and editing of the manuscript.

## Competing interests
The authors declare no competing interests.
