## [Peer Review File · Nature Communications]

Reviewers' comments:

Reviewer #1 (Remarks to the Author):

The authors performed a highly robust bioinformatic analysis of the putative LCT-T domain based on known sequences from a well-studied LCT. Using results from this analysis the authors performed a series of experiments to define the minimally functional translocation domain of *C. difficile* TcdB/TcdA. These are two somewhat distinct stories connected based on the bioinformatic analysis. The second part of the study is very informative and will provide a more detailed understanding of how TcdB and other LCTs translocate their enzymatic domains into the cytosol of the cell. The first part of the story, which identifies putative homologs to the T domain is interesting but still speculative in nature. I was a bit surprised that the authors did not functionally test any of the T domain homologs in the same manner that the T-domain of TcdB was analyzed. This would have provided more definitive evidences for the function of these proteins in the transport of effectors. The fact that the highest level of conservation was found in amino acids that had been shown to be important to T domain function may be enough to sway some readers, but isn't a substitute for experimentally testing one or more of the homologs. It is worth noting that the recent study in which BoNT was found in *Enterococcus faecium* demonstrated that the homolog cleaved VAMP-2 and SNAP25. That, in my opinion, has set the precedent for functionally testing putative homologs identified in these types of bioinformatic analyses.

1) There were several statements that were difficult to interpret. For example, the authors state the LCT-T homologs share an average of 18.6% identity with the TcdB translocase domain. What is the range of identity? Are there homologs with much higher identity that might raise the overall average of the group?

2) As I understand the data, no LCT-T were identified with T domain in the same gene as the putative effector. The supplemental data indicates the "effectors" were found in genes upstream of the protein with the LCT-T homology, but not as domains within the same putative protein. It seems necessary to purify the effector and T-domain and show some sort of function activity

3) I also found it difficult to understand the significance of the data in Figure 6. Isn't it predictable that the fragments will exhibit pore forming activity because the pore-forming region is present in the fragments. This section could be improved to help the readers better understand how this expands the understanding of the T-domain and pore formation.

4) Over 200,000 genomes were searched for T-domain homologs and 1,104 candidates were identified. This is suggested to be a remarkable number, but it's not clear to me why the authors thought this was an unexpectedly high number. Perhaps it is, but there is no similar analysis with other regions of the LCT proteins for relative comparisons. What is the result, for example, of a similar analysis with CROP repeat sequences? Or what is the outcome of a search with scrambled sequences of the T-domain? I appreciate the comparison with DT and BoNT translocase, but I don't

know if that addresses this issue. Again, without some sort of functional evidence for the T-domain homologs, it's difficult to put all of this in context.

Reviewer #2 (Remarks to the Author):

The manuscript by Orrell et al. identifies a translocation domain in TcdB, a large clostridial toxin in *C. difficile*, is also conserved in a number of species in a phylogenetically distant phylum of bacteria, proteobacteria. They used a bioinformatic approach (PSI-Blast) was used with the translocation domain of TcdB to identify additional proteins carrying this domain. This suggests this domain is more widespread and may assist in identifying additional new toxins previously unappreciated in these species (although this is not tested). They perform a series of truncation and biochemical experiments to show the "minimal" T-domain of TcdB that is capable of pore formation and translocation in a pH dependent manner. A studies of LCTs has been challenging and somewhat controversial in terms of receptor identification and other domain analysis, the defining of this translocation domain is of interest. However, the manuscript is heavily oriented towards the fact this domain is conserved in proteins (and organisms) that don't have LCTs and thus it was surprising that the authors do not experimentally test any of the proteobacteria T domains that were identified in the bioinformatic screen. Without experimental validation, it is unclear whether or not these domains serve the same function in proteobacteria, as suggested. Since the authors have come up with a heterologous system to test the truncated TcdB T domain, it seems logical that this system should be readily able to test a series of proteobacterial T domains to show they have the same function. Otherwise, the evidence (and the main focus of the paper) for this functional domain being widely distributed is largely circumstantial.

Specific comments.

1. The PSI-BLAST analysis is pretty tricky to use in this situation as the proteins identified using just the T-domain in the initial blast to build the PSSM for the second blast will include a lot of protein sequence that is unique from T-domains. Probably this is why 2 iterations were run. It might be useful to report how many initial T domains were found in the BLAST vs how many additional T-domains were found in the two PSI-BLAST iterations.

2. Given the scope of the number of genomes sequenced and the relatively limited number of bacterial genera that have this T domain, I think it is a bit overstated how remarkable the

distribution of this domain is in bacteria. It is interesting it is found heavily in *Pseudomonas* and *Vibrio*, but it is not a wide taxonomic distribution stated in the paper.

3. The newly identified T domains were noted to be near other well-known Gram-negative Type 1-4 secretion systems. My initial thought was these T domain proteins may be effectors that use these systems to translocate rather than just the T domain. Showing one or more of the T domains from proteobacteria are sufficient for translocation is thus needed to support some of the main conclusions of the paper.

4. In figure 4/supp. Fig 6 there is some data showing that only the 1500 and 1473 truncations are able to translocate as measured by viability via a cell rounding assay. S6 shows the ones that didn't work while Figure 4 reports the EC50 for cell viability for the two constructs that had activity. It would be more useful to have the %viability data of all of the constructs in S6, along with statistics.

5. Figure S3. In the *C. difficile* part of the figure, I think the first TcdB should be labeled TcdA

6. I found the supplementary table confusing. Please provide more information about the table and what the information means.

Reviewers' comments:

Reviewer #1 (Remarks to the Author):

The authors performed a highly robust bioinformatic analysis of the putative LCT-T domain based on known sequences from a well-studied LCT. Using results from this analysis the authors performed a series of experiments to define the minimally functional translocation domain of *C. difficile* TcdB/TcdA. These are two somewhat distinct stories connected based on the bioinformatic analysis. The second part of the study is very informative and will provide a more detailed understanding of how TcdB and other LCTs translocate their enzymatic domains into the cytosol of the cell. The first part of the story, which identifies putative homologs to the T domain is interesting but still speculative in nature. I was a bit surprised that the authors did not functionally test any of the T domain homologs in the same manner that the T-domain of TcdB was analyzed. This would have provided more definitive evidences for the function of these proteins in the transport of effectors. The fact that the highest level of conservation was found in amino acids that had been shown to be important to T domain function may be enough to sway some readers, but isn't a substitute for experimentally testing one of more of the homologs. It is worth noting that the recent study in which BoNT was found in *Enterococcus faecium* demonstrated that the homolog cleaved VAMP-2 and SNAP25. That, in my opinion, has set the precedent for functionally testing putative homologs identified in these types of bioinformatic analyses.

We thank the reviewer for their helpful comments and suggestions. Functionally testing the T domain homologs prior to our initial submission was something we had hoped for, however, we had run into several difficulties associated with expressing sufficient quantities of these sequences in *E.coli* – something that is challenging even for the LCT toxins. In the intervening time since submission, we were able to obtain large quantities of a distant homologue from *S. marcescens*. As you will see below, we have functionally characterized this homologue and are happy to include this new data in the revised manuscript (see details below).

1) There were several statements that were difficult to interpret. For example, the authors state the LCT-T homologs share an average of 18.6% identity with the TcdB translocase domain. What is the range of identity? Are there homologs with much higher identity that might raise the overall average of the group?

To clarify the comments about sequence identity among LCT translocases and the more distantly related LCT-like translocases, we have created a new supplementary figure visualizing sequence identity distributions. The figure legend is as follows:

Supplementary Figure 1. Comparison of detected clostridial LCT (a) and non-LCT (b) translocases to the TcdB translocase. The LCT translocases correspond to partial and complete translocase sequences from TcdA, TcdB, TcsL, TcsH, TpeL, and TcnA proteins, which possess greater sequence identity over a larger portion of the TcdB translocase compared to LCT-like translocases found outside of the clostridia.

2) As I understand the data, no LCT-T were identified with T domain in the same gene as the putative effector. The supplemental data indicates the “effectors” were found in genes upstream of the protein with the LCT-T homology, but not as domains within the same putative protein. It seems necessary to purify the effector and T-domain and show some sort of function activity

LCT-T homologs do have upstream domains to their T-domain, many of which are annotated and detailed in “LCT-T homologs occur in putative toxins with diverse effector types”. We appreciate that this may have been confusing, and have clarified this section in the text.

We have recombinantly produced an LCT-T homolog from *S. marcescens*, and have investigated both its effector and T-domain activity. The results are summarized in Supplementary Fig. 6. In text, we have included:

In order to corroborate the function of LCT-T homologs as toxins, we experimentally tested a homolog from Serratia marcescens (WP_073532240.1). In addition to having homology to the LCT-T domain, the S. marcescens putative toxin has a C80 peptidase (Supplementary Fig. 5a). Addition of a known allosteric activator of the C80 peptidase, Insp6, resulted in cleavage of the full length protein to yield two smaller fragments at ~63 kDa and ~25 kDa, which we speculate to be the effector region (residues ~1-600) and the C80 peptidase (~600-821), respectively (Supplementary Fig. 5b). When added to cells, S. marcescens toxin caused morphological changes (rounding) (Supplementary Fig. 5c), suggesting the protein, and by extension, the distant homologs in this study, do indeed function as a toxin by delivering a toxic effector into cells.

And

With the discovery of an evolutionarily conserved translocase region (herein referred to as ECT), we sought to test whether distant homologs containing the ECT retained essential translocation features. We again focused on the LCT-T homolog from S. marcescens, and tested its ability to form pores. In our dye release assay, the S. marcescens toxin did indeed form pores in a pH-dependent manner, forming pores at low pH, but not high pH (Supplementary Fig. 5d). Our results strongly support that the ECT confers pH-dependent pore formation and translocation activity to distant homologs, enabling homologs to act as pH-dependent protein translocases to deliver effectors into cells.

3) I also found it difficult to understand the significance of the data in Figure 6. Isn't it predictable that the fragments will exhibit pore forming activity because the pore-forming region is present in the fragments. This section could be improved to help the readers better understand how this expands the understanding of the T-domain and pore formation.

Based on the structure of the TcdA/TcdB T-domain at neutral pH, we would contend that it is far from predictable that fragments will exhibit pore formation and translocation activity. Additionally, no studies have shown that smaller fragments of the T-domain are functionally active. This significance of smaller T-domain fragments retaining functional activity is clarified in the text:

In context of the entire TcdB T-domain, the ECT extends from one end of the T-domain to the other, is a mixture of both helical and β -sheet content and generally does not look like an autonomous protein domain (Fig. 6a); further, no studies have shown that smaller fragments of the T-domain retain pore formation and translocation activity.

4) Over 200,000 genomes were searched for T-domain homologs and 1,104 candidates were identified. This is suggested to be a remarkable number, but it's not clear to me why the authors thought this was an unexpectedly high number. Perhaps it is, but there is no similar analysis

with other regions of the LCT proteins for relative comparisons. What is the result, for example, of a similar analysis with CROP repeat sequences? Or what is the outcome of a search with scrambled sequences of the T-domain? I appreciate the comparison with DT and BoNT translocase, but I don't know if that addresses this issue. Again, without some sort of functional evidence for the T-domain homologs, it's difficult to put all of this in context.

Excellent point. We performed a few analyses to support these claims. Of the 1104 LCT-like translocation domain sequences in the data set, 1023 sequences produce *E*-values less than $1e-5$ by comparison with TcdB using the FASTA package's search command, with 10,000 shuffled sequences to calculate statistics (reference: <https://www.pnas.org/content/85/8/2444.short>). This reaffirms the relationship between the TcdB translocation domain and translocation domain-like segments found in other species.

Second, when the other LCT domains are searched by PSI-BLAST, many more homologs are found. Queries of the TcdB glucosyltransferase domain (UniprotKB P18177, residues 1-565), the peptidase domain (residues 567-774), or the CROPS domain (residues 1815-2361) searched using two iterations of PSI-BLAST yield 5097, 3339, and >20,000 hits, respectively. Matches to these domains are more common than matches to the LCT translocase domain, although both the translocase and non-translocase components of LCTs are more common than the domains of other bacterial toxins (i.e., BoNTs and DTs).

However, considering this issue was raised independently by two reviewers, we have revised the manuscript to reduce these potential overstatements.

The concern regarding functional validation of T-domain homologues was brought up in reviewer comment #2. We agree that it is important to validate homolog function, and refer the reviewer to our supplementary figure 7, as detailed in comment 2.

Reviewer #2 (Remarks to the Author):

The manuscript by Orrell et al. identifies a translocation domain in TcdB, a large clostridial toxin in *C. difficile*, is also conserved in a number of species in a phylogenetically distant phylum of bacteria, proteobacteria. They used a bioinformatic approach (PSI-Blast) was used with the translocation domain of TcdB to identify additional proteins carrying this domain. This suggests this domain is more widespread and may assist in identifying additional new toxins previously unappreciated in these species (although this is not tested). They perform a series of truncation and biochemical experiments to show the "minimal" T-domain of TcdB that is capable of pore formation and translocation in a pH dependent manner. A studies of LCTs has been challenging and somewhat controversial in terms of receptor identification and other domain analysis, the defining of this translocation domain is of interest. However, the manuscript is heavily oriented towards the fact this domain is conserved in proteins (and organisms) that don't have LCTs and thus it was surprising that the authors do not experimentally test any of the proteobacteria T domains that were identified in the bioinformatic screen. Without experimental validation, it is unclear whether or not these domains serve the same function in proteobacteria, as suggested. Since the authors have come up with a heterologous system to test the truncated TcdB T domain, it seems logical that this system should be readily able to test a series of proteobacterial T domains to show they have the same function. Otherwise, the evidence (and the main focus of the paper) for this functional domain being widely distributed is largely circumstantial.

We thank the reviewer for their helpful comments and suggestions. As stated above and detailed below in (3), we have successfully produced and characterized a homologue to address this important point raised by both reviewers.

Specific comments.

1. The PSI-BLAST analysis is pretty tricky to use in this situation as the proteins identified using just the T-domain in the initial blast to build the PSSM for the second blast will include a lot of protein sequence that is unique from T-domains. Probably this is why 2 iterations were run. It might be useful to report how many initial T domains were found in the BLAST vs how many additional T-domains were found in the two PSI-BLAST iterations.

Using the version of the NR database from 2018-11-05 (the closest release prior to the date the sequences in our dataset were retrieved), 1216 of the 1573 sequences detected by PSI-BLAST yield *E*-values less than $1e-5$ with a single BLASTP search. However, only 856 of the 1104 sequences from the curated dataset produce *E*-values at that level, indicating that their similarity to TcdB is remote. However, as demonstrated by the sequence shuffling approach in Reviewer 1's fourth comment, the relationship of TcdB to the remote homologs is in most cases statistically significant and likely non-random.

2. Given the scope of the number of genomes sequenced and the relatively limited number of bacterial genera that have this T domain, I think it is a bit overstated how remarkable the distribution of this domain is in bacteria. It is interesting it is found heavily in *Pseudomonas* and *Vibrio*, but it is not a wide taxonomic distribution stated in the paper.

We agree and have revised the manuscript to reduce the emphasis placed on these claims.

3. The newly identified T domains were noted to be near other well-known Gram-negative Type 1-4 secretion systems. My initial thought was these T domain proteins may be effectors that use these systems to translocate rather than just the T domain. Showing one or more of the T domains from proteobacteria are sufficient for translocation is thus needed to support some of the main conclusions of the paper.

After significant effort and optimization, we have recombinantly produced an LCT-T homolog from *S. marcescens*, and have investigated both its effector and T-domain activity. The results are summarized in Supplementary Fig. 6. In text, we have included:

*In order to corroborate the function of LCT-T homologs as toxins, we experimentally tested a homolog from *Serratia marcescens* (WP_073532240.1). In addition to having homology to the LCT-T domain, the *S. marcescens* putative toxin has a C80 peptidase (**Supplementary Fig. 5a**). Addition of a known allosteric activator of the C80 peptidase, *Insp6*, resulted in cleavage of the full length protein to yield two smaller fragments at ~63 kDa and ~25 kDa, which we speculate to be the effector region (residues ~1-600) and the C80 peptidase (~600-821), respectively (**Supplementary Fig. 5b**). When added to cells, *S. marcescens* toxin caused morphological changes (rounding) (**Supplementary Fig. 5c**), suggesting the protein, and by extension, the distant homologs in this study, do indeed function as a toxin by delivering a toxic effector into cells.*

And

*With the discovery of an evolutionarily conserved translocase region (herein referred to as ECT), we sought to test whether distant homologs containing the ECT retained essential translocation features. We again focused on the LCT-T homolog from *S. marcescens*, and tested its ability to form pores. In our dye release assay, the *S. marcescens* toxin did indeed form pores in a pH-dependent manner, forming pores at low pH, but not high pH (Supplementary Fig. 5d). Our results strongly support that the ECT confers pH-dependent pore formation and translocation activity to distant homologs, enabling homologs to act as pH-dependent protein translocases to deliver effectors into cells.*

4. In figure 4/supp. Fig 6 there is some data showing that only the 1500 and 1473 truncations are able to translocate as measured by viability via a cell rounding assay. S6 shows the ones that didn't work while Figure 4 reports the EC50 for cell viability for the two constructs that had activity. It would be more useful to have the %viability data of all of the constructs in S6, along with statistics.

*Note, Supplementary Fig. 6 is now Supplementary Fig. 8

We have modified Supplementary Fig. 8 to address these reviewer's comments. We have provided another concentration (10 nM) for non-toxic constructs (a), and have provided the cell viability curves and protein synthesis inhibition curves (b, c), for the toxic constructs.

5. Figure S3. In the *C. difficile* part of the figure, I think the first TcdB should be labeled TcdA

We thank the reviewer for noticing this error, which we have now corrected.

6. I found the supplementary table confusing. Please provide more information about the table and what the information means.

In the previous version of Supplementary Table 1, hundreds of columns had been added to the table in error. We have now corrected this in a new version, as well as clarified the table legend.

REVIEWERS' COMMENTS:

Reviewer #1 (Remarks to the Author):

I appreciate the authors' efforts to demonstrate pore-forming function of a putative LCT-T domain. The new data dramatically improves the overall impact of the study and addresses my previous major concern with the report. I have no other concerns.

Reviewer #2 (Remarks to the Author):

The authors have responded well to both reviewers and the additional information with showing activity with *S. marcescens* addresses a key issue with both reviews. However, the information provided about supplementary figure 6 is not sufficient and some issues remain with the data as presented. My specific comments about this new data are listed below.

1. This seems like a key part of the story and it is unclear as to why this data would be considered supplementary. It should be part of the manuscript and considered in the discussion.
2. The amount of the *S. marcescens* toxin used in the assays seems quite high compared to the TcdB ECT studies. It would be nice to see a dose response for cell viability and fluorescence release to be sure these effects are specific (with the possibility of using mutants for confirmation).
3. The amount of dye released is much lower for the Sm toxin compared to the TcdB constructs tested.
4. There is nothing in the material and methods about the purification of the Sm toxin nor how it was tested in the in vitro assays. This data should be treated with the same level of detail as all other data (although I know supplemental data often isn't). In this case it seems important to the story as it at least provides one instance of toxin activity that is speculated by the authors. That is also why a more thorough characterization should be done.

REVIEWERS' COMMENTS:

Reviewer #1 (Remarks to the Author):

I appreciate the authors' efforts to demonstrate pore-forming function of a putative LCT-T domain. The new data dramatically improves the overall impact of the study and addresses my previous major concern with the report. I have no other concerns.

We thank the reviewer for helping improve the manuscript.

Reviewer #2 (Remarks to the Author):

The authors have responded well to both reviewers and the additional information with showing activity with *S. marcescens* addresses a key issue with both reviews. However, the information provided about supplementary figure 6 is not sufficient and some issues remain with the data as presented. My specific comments about this new data are listed below.

1. This seems like a key part of the story and it is unclear as to why this data would be considered supplementary. It should be part of the manuscript and considered in the discussion.

We agree and have added an entire new section to the main part of the manuscript with a new- and in fact, have included additional data that we recently collected, which further supports the functionality of Sm protein. Supplementary figure 6 is now main text figure 3.

2. The amount of the *S. marcescens* toxin used in the assays seems quite high compared to the TcdB ECT studies. It would be nice to see a dose response for cell viability and fluorescence release to be sure these effects are specific (with the possibility of using mutants for confirmation).

It is true that the absolute amount of Sm toxin required to induce cell-rounding was higher than is used for the LCT toxins; however, we feel that this is not unexpected and may reflect the fact that the specific cell types used to measure toxicity may not have high levels of its native receptor. Indeed, Sm protein may have greater activity against a different eukaryotic host. Nevertheless, to address this comment, we assessed additional human cells and found that human colonic carcinoma cells were more sensitive to the Sm toxin and have updated Fig. 3 to include dose-dependent cell rounding microscope images. To complement our cell rounding data and to show that the cell rounding is specific to the effector, we also performed transfection studies with the effector region. These data are included in Fig. 3.

With regards to the fluorescence dye release, we show that the toxin requires low pH to induce dye

release and the levels observed are comparable to that of the ADPR-[truncated TcdBT-domain]-DTR chimeras and full length TcdB (refer to comment #3).

3. The amount of dye released is much lower for the Sm toxin compared to the TcdB constructs tested.

The amount of dye release for the Sm toxin is actually not lower than the ADPR-[truncated TcdBT-domain]-DTR chimeras (Fig 5 - the dye release after 20 min is ~15-20%). We apologize if this was not clearer. We would also like to point out that full length TcdB also releases ~20% dye from HPTS/DPX loaded liposomes. TcdB dye release from HPTS/DPX liposomes has been published by Genisyuerek et. al. 2011 in *Molecular Microbiology*.

From our studies and others, full length (or chimeric) toxins release less dye than their T-domains alone. In our study, the TcdB ECT (i.e. 851-1473), releases more dye alone (Fig. 7) than in context of an effector and receptor-binding domain (Fig. 5). We believe this may be attributable to blockage of the pore by the effector, but look forward to investigating this further in future studies.

4. There is nothing in the material and methods about the purification of the Sm toxin nor how it was tested in the in vitro assays. This data should be treated with the same level of detail as all other data (although I know supplemental data often isn't). In this case it seems important to the story as it at least provides one instance of toxin activity that is speculated by the authors. That is also why a more thorough characterization should be done.

This was an oversight on our part. We thank the reviewer for catching this. We have updated the methods section with more technical information. We also provide more thorough characterization of the Sm toxin, as outlined in comment #2.